# Resistance to Mecillinam and Nine Other Antibiotics for Oral Use in *Escherichia coli* Isolated from Urine Specimens of Primary Care Patients in Germany, 2019/20

**DOI:** 10.3390/antibiotics11060751

**Published:** 2022-05-31

**Authors:** Michael Kresken, Yvonne Pfeifer, Florian Wagenlehner, Guido Werner, Esther Wohlfarth

**Affiliations:** 1Antiinfectives Intelligence GmbH, c/o Rechtsrheinisches Technologie- und Gründerzentrum, Gottfried-Hagen-Straße 60-62, 51105 Cologne, Germany; esther.wohlfarth@antiinfectives-intelligence.de; 2Rheinische Fachhochschule gGmbH, Schaevenstraße 1a-b, 50676 Cologne, Germany; 3Division Nosocomial Pathogens and Antibiotic Resistances, Department of Infectious Diseases, Robert Koch Institute, Burgstraße 37, 38855 Wernigerode, Germany; pfeifery@rki.de (Y.P.); wernerg@rki.de (G.W.); 4Clinic for Urology, Pediatric Urology and Andrology, Justus-Liebig-University Gießen, Rudolf-Buchheim-Straße 7, 35392 Giessen, Germany; florian.wagenlehner@chiru.med.uni-giessen.de

**Keywords:** *Escherichia coli*, uropathogen, outpatients, UTI, primary care, ESBL, mecillinam, fosfomycin, nitrofurantoin

## Abstract

Urinary tract infections (UTIs) are among the most common bacterial infections in humans. *Escherichia coli* is by far the leading cause of community-acquired UTIs. Pivmecillinam, the oral prodrug of the penicillin derivative mecillinam (amdinocillin), was re-introduced in Germany in March 2016 for first-line treatment of acute uncomplicated cystitis. This study aimed to evaluate the prevalence of resistance to mecillinam in comparison to nine other antibiotics used for oral treatment in *E. coli* urine isolates after the re-introduction of pivmecillinam. A total of 460 isolates were collected at 23 laboratories of clinical microbiology between October 2019 and March 2020. Forty-six isolates (10.0%) produced an extended-spectrum β-lactamase (ESBL) of the CTX-M family. Resistance to amoxicillin (43.3%) was most widespread, followed by resistance to trimethoprim-sulfamethoxazole (27.0%), amoxicillin-clavulanic acid (18.0%), cefuroxime (11.3%), and ciprofloxacin (11.1%). Twenty-four *E. coli* isolates (5.2%) were resistant to mecillinam. The concentrations of mecillinam needed to inhibit 50/90% of the ESBL-producing isolates and the remaining isolates were 1/4 mg/L and 0.5/4 mg/L, respectively. The findings support the recommendation to regard pivmecillinam as a first-line option for the treatment of uncomplicated lower UTIs.

## 1. Introduction

Urinary tract infections (UTIs) are among the most common bacterial infections in humans, affecting approximately 150–250 million patients globally per year [1]. *Escherichia coli* is the most frequent uropathogen, accounting for 70–80% of cases of acute uncomplicated lower UTI [2,3]. Further causative agents of uncomplicated lower UTI include *Klebsiella pneumoniae* (4–6%), *Staphylococcus saprophyticus* (4–6%), *Enterococcus* spp. (4–5%), *Proteus mirabilis* (2–4%), Group B streptococci (2–3%), and many others [2,3]. Increasing rates of resistance to oral standard antibiotics (e.g., aminopenicillins ± β-lactamase inhibitors, oral cephalosporins, fluoroquinolones, trimethoprim ± sulfamethoxazole) in *E. coli* and other members of the order Enterobacterales in the past 20–30 years have renewed interest in pivmecillinam for oral treatment of uncomplicated lower UTI.

Pivmecillinam is a prodrug of the “old” β-lactam antibiotic mecillinam, a 6 β-amidinopenicillanic acid derivative, which shows in vitro activity against many aerobic Gram-negative bacteria, including *E. coli* and other Enterobacterales order members. Its antibacterial activity derives from the ability to bind specifically to Penicillin Binding Protein-2 [4]. Mecillinam has been demonstrated to be relatively stable to hydrolysis by β-lactamases compared to other penicillins [5] and has been shown to offer clinically sufficient activity against most ampicillin-resistant *E. coli,* producing various types of β-lactamases, including AmpC enzymes, many extended-spectrum β-lactamases (ESBLs) and some carbapenemases [5,6,7,8,9].

Pivmecillinam was re-introduced in Germany in March 2016. It is currently available in several European, Asian and African countries, as well as in Canada, and received Qualified Infectious Disease Product designation from the Food and Drug Administration in the United States. In current national and international guidelines, pivmecillinam is among the first-line drugs recommended for empirical treatment of acute uncomplicated cystitis [10,11,12]. Dosing regimens, however, vary and include 400 mg either two times daily or three times daily for three days [10], three times daily for 3–5 days [11] and two times daily for five days [12]. The present study compared the in vitro activity of mecillinam to other orally administered antibiotics against *E. coli* isolated from urine specimens of primary care patients in Germany.

## 2. Results

### 2.1. Clinical Isolates and Patient Characteristics

A total of 460 *E. coli* urine isolates were collected. Each of the 23 laboratories provided 20 isolates, as requested. Three hundred and ninety-three (85.4%) and sixty-seven (14.6%) isolates were obtained from female and male patients, respectively. Among the group of females, 30 isolates (6.5%) were from young patients (<18 years), 189 (41.1%) were from females aged 18–65 years, and 174 (37.8%) were from elderly females (>65 years). The median (interquartile range) of patients’ age was 63 (45–78) years.

Forty-six isolates (10.0%) produced an ESBL, nineteen (41.3%) and 18 (39.1%) of which were obtained from females aged 18–65 years and >65 years, respectively. Twenty-five isolates produced one or more other β-lactamase (TEM (*n* = 12), DHA (*n* = 1), OXA-1 group (*n* = 10), OXA-48-like (*n* = 2)). Subsequent polymerase chain reaction (PCR) typing and Sanger sequencing of the two *bla*_OXA-48_-like positive isolates revealed the gene *bla*_OXA-244_ present in a clonal subgroup of *E. coli* sequence type (ST)38 that has been recently reported from various European countries, including Germany [13]. CTX-M-type ESBLs were present in all ESBL-producing isolates. Thirty isolates were positive for CTX-M group 1, 15 for CTX-M group 9 and one isolate for CTX-M-8. Nineteen (41.3%) and four (8.7%) CTX-M-producing isolates belonged to the *E. coli* subgroups O25b-ST131 and O16-ST131, respectively. Furthermore, two isolates were resistant to third-generation cephalosporins and AmpC producers only, with enzymes CMY and DHA, and one isolate produced DHA plus TEM.

### 2.2. Antimicrobial Susceptibility of Clinical Isolates

The range of minimum inhibitory concentrations (MICs), the MICs inhibiting 50% and 90% of the isolates (MIC 50, MIC 90), and the number and percent of susceptible (S + I, see Section 2 for definitions of S and I) and resistant isolates for mecillinam and the comparative agents are shown in Table 1.

Resistance to amoxicillin (43.3%) was most widespread, followed by resistance to trimethoprim-sulfamethoxazole (27.0%), amoxicillin-clavulanic acid (18.0%), cefuroxime (11.3%) and ciprofloxacin (11.1%). Resistance to fosfomycin was confirmed in 34 (7.4%) isolates, and “high-level” resistance to amoxicillin-clavulanic acid (MIC > 32 mg/L being the relevant breakpoint for isolates from patients with uncomplicated UTI) was detected in 26 isolates (5.7%). Five (1.1%) isolates were resistant to nitrofurantoin. Twenty-four *E. coli* isolates (5.2%) were resistant to mecillinam. The MIC distribution data of mecillinam are presented in Table 2. Mecillinam, at the breakpoint of 8 mg/L, inhibited 42/46 (91.3%) ESBL-producing isolates and 394/414 (95.2%) non-ESBL isolates (odds ratio 0.53, 95% confidence interval (C.I.) 0.17–1.63).

Of the 460 isolates, 226 (49.1%) were fully susceptible to the eight drug classes/subclasses tested: penicillins (ATC code J01CA: amoxicillin, mecillinam), penicillins + β-lactamase inhibitors (J01CR: amoxicillin-clavulanic acid), second-generation cephalosporins (J01DC: cefuroxime), third-generation cephalosporins (J01DD: cefixime, cefpodoxime), fluoroquinolones (J01MA: ciprofloxacin), folate pathway inhibitors (J01EE: trimethoprim-sulfamethoxazole), phosphonic acids (J01XX01: fosfomycin) and nitrofurans (J01XE: nitrofurantoin). Sixty-seven isolates (14.6%) were resistant to one drug class, seventy (15.2%) to two drug classes, and ninety-seven isolates (21.1%) met the criterion of multidrug resistance (≥three drug classes), but none were resistant to seven or all eight drug classes. Mecillinam at 8 mg/L inhibited 84/97 (86.6%) multidrug-resistant isolates, including nine isolates that were resistant to six drug classes.

### 2.3. Origin and Characterization of Mecillinam-Resistant Isolates

The 24 mecillinam-resistant isolates (MIC > 8 mg/L) were isolated at 15 laboratories. Seven laboratories each found one and two isolates, respectively, and one laboratory found three mecillinam-resistant isolates. Twenty-one mecillinam-resistant *E. coli* isolates were detected among the 393 isolates from females (5.4%), while three mecillinam-resistant isolates were detected among the 67 isolates from males (4.5%) (odds ratio 1.20, 95% C.I. 0.35–4.16). Further, resistance to mecillinam was observed in six of the 86 isolates from women aged 18–45 years (7.0%) as compared to 16 resistant strains in the 340 isolates from women aged > 45 years (4.7%) (odds ratio 1.52, 95% C.I. 0.58–4.00).

Resistance patterns of mecillinam-resistant isolates are displayed in Table 3. All but one of the mecillinam-resistant isolates showed cross-resistance to amoxicillin and amoxicillin-clavulanic acid (each 95.8%), while 13 (54.2%), five (20.8%), five (20.8%), four (16.7%), and three (12.5%) were additionally resistant to trimethoprim-sulfamethoxazole, cefuroxime, cefpodoxime, cefixime, and ciprofloxacin, respectively. Two mecillinam-resistant isolates were also resistant to fosfomycin, and one was also resistant to nitrofurantoin, while one was susceptible to all other drugs tested. Twenty mecillinam-resistant isolates were ESBL-negative, three harbored a *bla*_CTX-M_ group 1 gene plus *bla*_TEM_ (collected at two different sites) and one isolate of the clonal subgroup *E. coli* O25b-ST131 harbored a *bla*_CTX-M_ group 9 gene plus *bla*_OXA-244_. The mecillinam concentrations to inhibit amoxicillin-resistant isolates were higher than those to inhibit amoxicillin-susceptible isolates, with MIC 50 and MIC 90 values of 2 mg/L and 16 mg/L, respectively, for amoxicillin-resistant isolates, and 0.25 mg/L and 0.5 mg/L, respectively, for amoxicillin-susceptible isolates. The highest MIC 90 of mecillinam (>32 mg/L) was calculated for the 26 isolates showing “high-level” resistance to amoxicillin-clavulanic acid (Table 2).

## 3. Discussion

The present surveillance study involving 23 clinical microbiological laboratories across Germany investigated the occurrence of resistance to mecillinam and other oral antibiotics among 460 *E. coli* isolates obtained from urine samples of outpatients. The vast majority of isolates (>85%) were obtained from women, as expected. The finding that 43% of the isolates were resistant to amoxicillin and 27% to trimethoprim-sulfamethoxazole compared well with the results of a previous nationwide study performed in 2010 by our group [14]. In contrast, resistance to ciprofloxacin significantly decreased from 19.8% in 2010 to 11.1% in 2019 (difference 8.8%, 95% C.I. 4.2–13.3%). The decrease in resistance may be related to the reduction in the consumption of fluoroquinolones from 1.53 (2010) to 0.63 (2019) defined daily doses per 1000 inhabitants per day in the German primary care sector [15]. In this context, it is important to point out that in October 2018, the Pharmacovigilance Risk Assessment Committee of the European Medicines Agency recommended reducing the use of fluoroquinolones and quinolones after reviewing the adverse and potentially long-lasting side effects reported with these drugs [16].

Unlike resistance to fluoroquinolones, we observed no significant change in the occurrence of ESBL-producing *E. coli*. The ESBL rate in this study (10%) was even slightly higher than the rate found in 2010 (8%). Further, as in the study performed in 2010, *E. coli* O25b-ST131 was identified as the predominant ESBL-producing clonal subgroup [14].

The present study found 5.2% of the *E. coli* isolates to be mecillinam-resistant. This figure is significantly higher than the resistance rate of 2% (difference 3.2%, 95% C.I. 0.8–5.8%) that we noticed in a study conducted prior to the introduction of the first pivmecillinam product in Germany [17] but is in line with the resistance rate reported for *E. coli* urine isolates from the primary care sector in Denmark (5.3% in 2019) [18]. For comparison, the proportion of mecillinam-resistant *E. coli* among invasive and urine isolates from hospital patients in Denmark was 14% and 8.1%, respectively, while the Swedish resistance surveillance system reported an overall mecillinam resistance rate of 4.8% for *E. coli* urine isolates in both 2019 and 2020 [19,20]. In Sweden, the proportion of mecillinam-resistant strains in all *E. coli* urine isolates in the period 1996–2009 was almost always 1–2% and subsequently 4–5% per year. The number of pivmecillinam prescriptions in Sweden increased from <20 prescriptions per 1000 inhabitants before 2004 to >30 in 2008 and has remained almost constant thereafter, indicating a possible connection between consumption and frequency of resistance [20,21,22]. The Northern Dimension Antibiotic Resistance Study comprising medical laboratories in Finland, Germany, Latvia, Poland, Russia and Sweden investigated the level of antimicrobial resistance among *E. coli* urine isolates from female outpatients (aged 18–65 years) with symptoms of uncomplicated UTI [23]. The overall resistance rate to mecillinam was 4.1%, though considerable differences between countries were evident, ranging from 0% among 111 isolates from seven sites (Stockholm region) in Sweden to 10.5% among 95 isolates from a single site (Silesian Voivodeship) in Poland. Unfortunately, Germany was missing susceptibility data for mecillinam. Overall non-susceptibility rates in that study determined for ampicillin (39.6%), trimethoprim-sulfamethoxazole (22.4%), ciprofloxacin (15.1%), cefuroxime (9.6%) and nitrofurantoin (1.2%) were comparable to those found in the present study. Likewise, the overall ESBL rate in that study (8.7%) approximated that in our study (10.0%) [23].

There are some limitations of the present study. The study did not include the type and severity of the UTI from which the urine *E. coli* isolates were obtained. Furthermore, the study was based on samples taken during primary care. It has been demonstrated that laboratory-based surveillance studies tend to overestimate the prevalence of resistance as they usually include disproportionately more isolates from patients with complicated UTIs and risk factors such as previous antimicrobial treatment failure [24,25], compared with patient-based studies such as ARESC [2]. The rate of mecillinam resistance in isolates from patients with acute uncomplicated cystitis, which is the approved indication of pivmecillinam in Germany, may thus be lower than the rate of 5.2% reported here.

Moreover, the importance of mecillinam resistance for the therapy of UTIs has been questioned from a clinical point of view. Thulin et al. reported that inactivation of the *cysB* gene is the major cause of mecillinam resistance in clinical isolates of *E. coli* [26]. The *cysB* gene encodes for the CysB protein, which is the major positive regulator of the cysteine biosynthesis pathway. Thulin and Andersson showed that *cysB* mutations led to increased levels of the proteins PBP1B, LpoB, and FtsZ, which are known to be involved in peptidoglycan biosynthesis [27]. CysB-related mecillinam resistance, however, was only expressed in growth media with low concentrations of cysteine (including Mueller–Hinton broth) and *cysB* mutants returned to mecillinam susceptibility when the media were supplemented with cysteine. Many mecillinam-resistant *E. coli cysB* mutants also showed phenotypic susceptibility in urine, whereby the degree of reversion to susceptibility was correlated with osmolality such that low osmolality favored susceptibility [26,28]. These experiments are still pending with the mecillinam-resistant isolates found in the present study. However, at least in the amoxicillin-susceptible isolate of the present study, resistance to mecillinam must be due to a different mechanism.

## 4. Materials and Methods

### 4.1. Bacterial Strains

Isolates were collected during a laboratory-based surveillance study carried out between October 2019 and March 2020 by the Study Group ‘Antimicrobial Resistance’ of the Paul-Ehrlich-Society for Infection Therapy. Twenty-three laboratories of clinical microbiology throughout Germany were requested each to collect 20 consecutive, non-duplicate *E. coli* urine isolates from primary care outpatients. Additional information collected with each isolate was the isolation date and age and gender of the patients. At the end of the collection period, all isolates were shipped to a central laboratory (Antiinfectives Intelligence, Cologne, Germany) for further analyses.

### 4.2. Confirmation of Species Identification and Susceptibility Testing

Species identification was confirmed using a MALDI Biotyper (Bruker Daltonik, Bremen, Germany).

Mecillinam powder was purchased from TOKU-E (Bellingham, WA, USA). MICs of the antibacterial agents were determined according to the methods described in the International Organization for Standardization (ISO) document 20776-1 [29] using concentrations derived from serial two-fold dilutions indexed to the basis 2. The agar dilution method was used to determine the mecillinam MICs (0.03–32 mg/L), as described in the Clinical Laboratory Standard Institute (CLSI) document M7-A10 [30]. Agar plates were prepared in-house. The broth microdilution procedure was employed on the other antibacterial agents tested [29]. Broth microdilution MICs were determined by applying a commercially ready-to-use test system that uses vacuum-dried antibiotics in 96-well microtiter plates (MICRONAUT-S; Merlin Diagnostika, Bornheim, Germany). Comparative orally administered antibiotics tested were amoxicillin (0.5–32 mg/L), amoxicillin-clavulanic acid (clavulanic acid concentration fixed at 2 mg/L; 0.5/2–32/2 mg/L), cefuroxime (administered orally as cefuroxime axetil; 0.12–32 mg/L), cefixime (0.03–4 mg/L), cefpodoxime (0.06–4 mg/L), ciprofloxacin (0.06–8 mg/L), trimethoprim-sulfamethoxazole (ratio 1:19; 0.25/4.75–16/304 mg/L), nitrofurantoin (16–256 mg/L), and fosfomycin (supplemented with 25 mg/L glucose-6-phosphate; 1–1024 mg/L). Resistance to fosfomycin was confirmed by the agar dilution method [30]. If the fosfomycin MIC value determined with the agar dilution method was different from that of the broth microdilution procedure, the result of the agar dilution method was used for the analysis of the data. Further, cefotaxime (0.12–16 mg/L) and ceftazidime (0.25–32 mg/L), alone and in combination with clavulanic acid (concentration fixed at 4 mg/L), were tested for ESBL detection (see below).

The accuracy of susceptibility testing was evaluated using quality control strains *E. coli* ATCC 25922 and *Pseudomonas aeruginosa* ATCC 27853. The size of the inoculum (broth microdilution, 2 × 10^5^–8 × 10^5^ colony-forming units (CFU)/mL; agar dilution, 1 × 10^4^ CFU per spot) was verified by counting CFUs. This was conducted for all tests performed with the reference strains and for 10% of the tests performed with the clinical isolates. Isolates were defined as S (susceptible, standard dosing regimen), I (susceptible, increased exposure), or R (resistant) to antimicrobial agents in accordance with the species-related clinical breakpoints approved by the European Committee on Antimicrobial Susceptibility Testing (EUCAST, version 12.0). The breakpoints for oral administration were used, if defined [31].

Multidrug resistance was defined as resistance to at least three of the eight antibacterial drug classes/subclasses (see Section 2.2 for details).

### 4.3. Phenotypic and Molecular Detection of AmpC Enzymes and ESBLs

Isolates with MICs > 1 mg/L for cefotaxime and/or ceftazidime were tested for AmpC or ESBL production. Phenotypic ESBL detection was based on the testing of both cefotaxime and ceftazidime, alone and in combination with clavulanic acid (see above), according to the broth microdilution procedure described by EUCAST [32] and the CLSI [33]. Isolates exhibiting a higher MIC (≥three two-fold dilution steps) of cefotaxime and/or ceftazidime alone compared with the MIC of the respective combination with clavulanic acid were considered ESBL-positive. All other isolates were suspected to have produced an AmpC enzyme.

Isolates were then further characterized by PCR and, in part, Sanger sequencing. PCR screening for β-lactamase genes (*bla*_TEM_-type, *bla*_SHV_-type, *bla*_CTX-M-1-2-8-9-25_ group, *bla*_OXA-1_-group, *bla*_OXA-48_-like, *bla*_CMY_-like, *bla*_DHA_-like) was conducted using primers and conditions that have been described previously [34]. *E. coli* isolates harboring genes encoding ESBLs or AmpC β-lactamases were further screened by PCR for the presence of the rfbO25b and rfbO16 genes that are associated with isolates belonging to the clonal subgroups *E. coli* O25b-ST131 and O16-ST131, respectively [35].

### 4.4. Data Processing and Statistical Evaluation

Data were processed using Microsoft Excel 2019 (Microsoft Corp., Redmond, WA, USA). The strength of association between two events was quantified by the odds ratio. The 95% C.I.s were constructed using the Newcombe–Wilson method without continuity correction.

## 5. Conclusions

Our findings and the discovery that mecillinam resistance in *E. coli* is often conditional suggest that clinical resistance to mecillinam has been low in Germany, supporting the current recommendation to regard pivmecillinam as a first-line option for the treatment of uncomplicated cystitis. However, regular monitoring of mecillinam resistance and the investigation of underlying mechanisms are imperative in order to verify the status of pivmecillinam as a first-line medication.

## Figures and Tables

**Table 1 antibiotics-11-00751-t001:** In vitro activity of mecillinam and nine comparator agents against 460 *E. coli* urine isolates.

Antimicrobial Agent	MIC 50 (mg/L)	MIC 90 (mg/L)	MIC Range (mg/L)	%-S	%-I	%-R (95% C.I.)
Mecillinam oral (uUTI)	0.5	4	0.06–>32	94.8	–	5.2 (3.2–7.2)
Amoxicillin	4	>32	≤0.5–>32	56.7	–	43.3 (38.9–47.8)
Amoxicillin-clavulanic acid	4	16	≤0.5–>32	82.0	–	18.0 (14.5–21.5)
Amoxicillin-clavulanic acid (uUTI) ^1^	4	16	≤0.5–>32	94.3	–	5.7 (3.6–7.8)
Cefuroxime oral (uUTI)	4	>32	≤0.12–>32	88.7	–	11.3 (8.4–14.2)
Cefpodoxime (uUTI)	0.5	>4	≤0.06–>4	88.9	–	11.1 (8.2–14.0)
Cefixime (uUTI)	0.25	4	≤0.03–>4	89.3	–	10.7 (7.9–13.5)
Ciprofloxacin	≤0.06	8	≤0.06–>8	86.3	2.6	11.1 (8.2–14.0)
Trimethoprim-sulfamethoxazole ^2^	≤0.25	>16	≤0.25–>16	72.2	0.9	27.0 (22.9–31.1)
Fosfomycin (uUTI)	2	8	≤1–256	92.6	–	7.4 (5.0–9.8)
Nitrofurantoin (uUTI)	≤16	32	≤16–>256	98.9	–	1.1 (0.1–2.1)

S—susceptible, standard dosing regimen; I—susceptible, increased exposure; R—resistant; uUTI—uncomplicated urinary tract infection. EUCAST clinical breakpoints set for isolates from patients with uUTI were applied. ^1^ Amoxicillin-clavulanic acid susceptible, MIC ≤ 32 mg/L; resistant, MIC > 32 mg/L. ^2^ Trimethoprim-sulfamethoxazole in the ratio 1:19. MICs are expressed as the trimethoprim concentration.

**Table 2 antibiotics-11-00751-t002:** MIC distribution data for mecillinam against 460 *E. coli* urine isolates with different resistance phenotypes.

Phenotype	*n*	MIC (mg/L)											
		≤0.03	0.06	0.12	0.25	0.5	1	2	4	8	16	32	>32
ESBL-negative	414		7	23	163	85	41	47	23	5	6	9	5
ESBL-positive	46				4	11	10	14	3		1	1	2
AMX-susceptible	261		7	23	155	68	4	2	1		1		
AMX-resistant	199				12	28	47	59	25	5	6	10	7
AMC-susceptible	377		7	23	163	90	39	40	13	1	1		
AMC-resistant	83				4	6	12	21	13	4	6	10	7
AMC-resistant (HL)	26				4	1	4	5	3		1	2	6
Total	460		7	23	167	96	51	61	26	5	7	10	*7*

The underlined numbers indicate the MIC 50/90 values. The solid vertical line indicates the EUCAST clinical breakpoint defined for mecillinam susceptibility. ESBL—extended-spectrum β-lactamase; AMX—amoxicillin; AMC—amoxicillin-clavulanic acid; HL—“high-level” resistance (AMC MIC > 32 mg/L).

**Table 3 antibiotics-11-00751-t003:** Resistance patterns of the 24 mecillinam-resistant *E. coli* urine isolates.

Resistance Pattern	*n*
MEC, AMX, AMC	7
MEC, AMX, AMC, TRS	4
MEC, AMX, AMC, CXM, CFI, CPD, TRS	3
MEC, AMX, AMC, CIP, TRS	3
MEC, AMX, AMC, TRS, FOS	2
MEC, AMX, AMC, CXM, CFI, CPD	1
MEC, AMX, AMC, CPD, TRS	1
MEC, AMX, AMC, CXM	1
MEC, AMX, AMC, NFT	1
MEC	1

MEC—mecillinam; AMX—amoxicillin; AMC—amoxicillin-clavulanic acid; CFI—cefixime; CPD—cefpodoxime; CXM—cefuroxime; CIP—ciprofloxacin; NFT—nitrofurantoin; FOS—fosfomycin; TRS—trimethoprim-sulfamethoxazole.

## Data Availability

Not applicable.

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
