# Peer review of "Resistance to Mecillinam and Nine Other Antibiotics for Oral Use in Escherichia coli Isolated from Urine Specimens of Primary Care Patients in Germany, 2019/20"

_antibiotics, 2022, doi:10.3390/antibiotics11060751_

Round 1
Reviewer 1 Report
The manuscript "Resistance to mecillinam and nine other antibiotics for oral use in Escherichia coli isolated from urine specimens of primary care patients in Germany, 2019/20" is well written and consistent, however, I have some suggestions that need to be adressed:
- Line 48: "Mecillinam has been demonstrated to be relatively stable..." - Please explain in the text what does it mean. Stable in what sense?
- In Materials and Methods the authors describe performing PCR in order to screen for few beta-lactamase genes. However, only one (blaOXA-244) is mentioned in the results section and the reader is not provided with the information about other genes presence. Please complete this information in the text.
- Some of concentrations ranges in the Table 1 seem inaccurate and did not allow to assess the exact MIC value. Please explain why follow-up studies were not performed to determine MIC concentrations that were below or above the values determined by ready-to-use tools.
- Why the agar dilution method was chosen for mecillinam and the broth microdilution method for other drugs? Couldn't mecillinam be tested by BMD as well?
- Lines 255-258: It is not clear whether all agents were tested by both (agar dilution and broth microdilution) methods or not. Please clarify it in the text (lines 239-260).
- Why the agar dilution method was chosen as conclusive? (line 257).
- The data from the section 2.1 (lines 64-69) could be presented in a more "graphic" way.
Author Response
Dear Reviewer,
Please see attachment for our responses to your comments.
Kind regards,
Michael Kresken

Reviewer 2 Report
- The English need improvement since there are some grammatical and syntax errors in the manuscript. For example,
- in line number 66, the words “Median” may be as “The median”;
- in line number 166, “found out” as “found”;
- in line number 224, “mechanisms is” as “mechanisms are”;
- in line number 233, “isolation” as “the isolation”;
- in line number 247, “applying” as “by applying”.
The grammar mistakes which are not mentioned here are also to be checked and corrected properly.
- There are some typing mistakes as well, and authors are advised to carefully proofread the text. For example,
- in line number 19, the words “community acquired” may be as “community-acquired”;
- in line number 19 and 31, “UTI” as “UTIs”;
- in line number 80, “third generation” as “third-generation”;
- in line number 269, “colony forming” as “colony-forming”.
The typos not mentioned here are also to be checked and corrected properly.
- Check the abbreviations throughout the manuscript and introduce the abbreviation when the full word appears the first time in the text and then use only the abbreviation (For example, PCR, DHA, TEM, MIC, etc.). And it should be in both abstract as well as in the remaining part of the manuscript. Make a word abbreviated in the article that is repeated at least three times in the text, not all words need to be abbreviated.
- The intrudouction part appears less informative about the urinary tract infections, thus this section should be indicated as detailed to understand the manuscript in clear. Hence, the authors are encouraged to include what are all the other organisms that cause urinary tract infections. And also the recent data related to prevalence may be given since the author cited only 2015 data.
- In materials and methods, under data processing and statistical evaluation, the version of Excel may be included.
- The authors may improve the discussion of their work by focusing on the present findings and introducing other authors who also worked with the same or other studies with recent references shortly.
- In conclusion, it is highly recommended to include weaknesses of the study and potential future research goals.
Author Response

(The authors gave the same response as above.)

Round 2
Reviewer 1 Report
Dear Authors,
Thank you for clarifying all the issues. In the Table 1 I misunderstood the meaning of "MIC range" and I thought it described the tested concentrations, not the assessed ones.
I have no further comments.